# Factors associated with uptake of breast and cervical cancer screening among Nepalese women: Evidence from Nepal Demographic and Health Survey 2022

Bipul Lamichhane[1]*, Bikram Adhikari[1], Lisasha Poudel[2], Achyut Raj Pandey[1], Sampurna Kakchhapati[1], Saugat Pratap K. C.[1], Santosh Giri[1], Bishnu Prasad Dulal[1], Deepak Joshi[1], Ghanshyam Gautam[1], Sushil Chandra Baral[1]

1 Research and Development Department, HERD International, Lalitpur, Nepal, 2 Central Department of Public Health Institute of Medicine, Tribhuvan University, Kathmandu, Nepal

☯ These authors contributed equally to this work.
* lamichhaneipul9@gmail.com

**Data Availability Statement:** The data are publicly available in the official website of The DHS

## Abstract

Breast cancer screening (BCS) and cervical cancer screening (CCS) are integral parts of initiatives to reduce the burden associated with these diseases. In this context, we aimed to determine factors associated with BCS and CCS uptake among Nepalese women aged 30 to 49 years using data from the Nepal Demographic Health Survey (NDHS) 2022. We performed a weighted analysis to account complex survey design of the NDHS 2022. We employed univariable and multivariable logistic regression to determine factors associated with the uptake of BCS and CCS and results were presented as crude odds ratio and adjusted odds ratio (AOR) along with 95% confidence interval (CI). The uptake of BCS and CCS among Nepalese women aged 30 to 49 years were 6.5% and 11.4% respectively. Women from Terai compared to mountain region (AOR = 0.54, 95%CI: 0.31, 0.93) and those engaged in agriculture compared to non-working (AOR = 0.59, 95%CI: 0.42, 0.82) women had lower odds of BCS uptake. Conversely, Dalit women compared to Brahmin/Chhetri (AOR = 2.08, 95%CI: 1.37, 3.16), and women with basic (AOR = 1.49, 95%CI: 1.04, 2.13), secondary (AOR = 1.96, 95%CI: 1.33, 2.88), and higher education (AOR = 2.80, 95%CI: 1.51, 5.19) compared to those with no education had higher odds of BCS uptake. Women from rural areas (AOR = 0.76, 95%CI: 0.61, 0.96), and those living in Bagmati (AOR = 2.16, 95% CI: 1.44, 3.23) and Gandaki (AOR = 2.09, 95%CI: 1.40, 3.14) provinces had higher odds of CCS uptake compared to their urban counterparts and those living in Koshi province, respectively. The odds of CCS increased with age (AOR = 1.06, 95%CI: 1.04, 1.08). Women with secondary education (AOR = 1.47, 95%CI: 1.06, 2.04) had higher odds of CCS uptake compared to those without education. Similarly, married women (AOR = 8.24, 95%CI: 1.03, 66.21), and those with health insurance (AOR = 1.41, 95%CI: 1.08, 1.83) had higher odds of CCS. In conclusion, the uptake of both BCS and CCS was relatively poor among Nepalese women indicating a need for targeted and tailored intervention to increase BCS and CCS uptake.

Program (https://dhsprogram.com/data/dataset/
Nepal_Standard-DHS_2022.cfm?flag=0).

**Funding:** The authors received no specific funding
for this work.

**Competing interests:** The authors have declared
that no competing interests exist.

## Introduction

Cancer constitutes a significant public health concern and stands among the top contributors to mortality, responsible for almost 10 million deaths worldwide in 2020 [1]. In the same year, breast cancer affected 2.3 million women, resulting in 685,000 deaths worldwide making it the most prevalent type of cancer [2]. Meanwhile, cervical cancer is ranked as the fourth most common cancer among women on a global scale, with approximately 604,000 new cases and 342,000 deaths reported in 2020. Notably, low- and middle-income countries (LMICs) accounted for about 90% of the deaths caused by cervical cancer observed globally during that year [3].

The age-standardized incidence rate of breast cancer in 2017 was 11.5 per 100000, with a mortality rate of 6.8 per 100,000 women [4]. Similarly, the age-standardized incidence rate of cervical cancer in 2020 was estimated to be approximately 16.4 per 100,000 women, with a mortality rate of 11.1 per 100,000 women [1]. Mortality rates for breast and cervical cancers continue to remain a concern in LMICs particularly due to limited access to comprehensive treatment centers, and lack of awareness among the public.

Breast and cervical cancer screening are crucial strategies in preventing and detecting cancers at early stages. Mammography, clinical breast examination (CBE), and breast self-examination (BSE) are the three commonly used screening tests for breast cancer. Mammography is the recommended standard globally, but for low-resource settings, [5] CBE and BSE are also suggested due to its cost-effectiveness [6–8]. In Nepal, screening methods for cervical cancer include Papanicolaou (PAP) test, human papilloma virus (HPV) testing, and visual inspection with acetic acid. Early detection of breast and cervical cancers not only save lives but also preserve quality of life and reduces the financial burden associated with advanced-stage cancer treatment [9]. However, delayed detection of breast and cervical cancers remains a common issue, particularly in LMICs like Nepal [10, 11]. A significant proportion of women in these countries seek medical treatment only when the cancer has reached advanced stages [12]. For example, a clinical study in Nepal reported that breast cancer patients often sought treatment at advanced stages, with average tumor sizes ranging from two to five centimeters [13]. This delay results in larger tumors, complicating the treatment and leading to higher mortality rates within Nepal. The lack of universal health coverage adds a significant financial burden on affected individuals and their families [14].

A scoping review states that LMICs including Nepal have inadequate capacity to provide accessible breast and cervical cancer care screening and cancer care. In addition, Nepalese women had low levels of knowledge about breast and cervical cancer, and screening practices for breast and cervical cancer were inadequate among Nepalese women due to sociocultural, geographic, or financial barriers [15]. Supply-side factors contributing to low rates of screening include lack of skilled healthcare personnel, inadequate availability of equipment, challenges in sample transportation, and low number of laboratories with qualified staff. Also, the high cost associated with PAP test in Nepal, makes it unaffordable for the majority of women in Nepal [16].

Nepal's national guidelines for cervical cancer screening (2010) recommend screening with visual inspection with acetic acid every 5 years for women in the age range of 30–60 years [17]. The goal was set to target at least 50% of women aged 30–60 years in 2010 which was later revised to 70% in 2017 [17]. Though early screening of breast cancer serves as a leading role in reducing the burden of breast cancer mortality, Nepal has no national and regional breast cancer screening program [18]. Additionally, there has been no initiation of population-based breast cancer screening, [19] despite the American Cancer Society recommending that women with an average risk of breast cancer should start annual mammography screening at age 40 [20].

In the context of Nepal, social determinants play a crucial part in awareness and utilization of health services including breast and cervical cancer screening. Various factors limit Nepalese women's participation in breast cancer screening (BCS) and cervical cancer screening (CCS), including cultural taboos, stigma, low health literacy, financial barriers, geographic challenges, and gender inequality. This is further exacerbated by economic and geographical constraints that make accessing screening services difficult, especially for lower-income individuals and those living in remote areas. Gender disparities also affect women's autonomy in healthcare decision-making.

Currently, there are limited studies to determine the status of uptake of BCS and CCS among Nepalese women and their associated factors. Thus, this study aimed to determine the status of uptake of BCS and CCS and associated factors among Nepalese women, using the secondary data from Nepal Demographic and Health Survey (NDHS) 2022 [21]. This study contributes to a better understanding of the status of breast and cervical cancer screening uptake in Nepal that can inform policymakers in the development of policies and targeted interventions to improve screening uptake.

## Methods

### Study design

We analyzed secondary data from NDHS 2022, a nationally representative survey [21].

### Study setting

Nepal is a landlocked country located in South Asia occupying an area of 147, 516 km$^2$. The country has seven administrative provinces, consisting of 77 districts, 6 metropolitan cities, 11 sub-metropolitan cities, 276 urban municipalities, and 460 rural municipalities distributed across three ecological belts—mountains, hills, and Terai. Based on Census 2021, the total population of Nepal is 29,164,578 of whom 51.1% are females [22]. Nepal has an overall human development index (HDI) of 0.587 with the HDI of rural and urban settings being 0.647 and 0.561, respectively [23].

### Sample size and sampling technique

The sample size and sampling technique of NDHS 2022 is described elsewhere [21]. In brief, the study used implicit stratification with proportional allocation at each lower administrative level to achieve a representative sample. This involved sorting the sampling frame within each stratum before selecting samples based on probability-proportional-to-size at the first stage. In the first stage of sampling, 476 (248 from urban and 228 from rural) primary sampling units (PSUs) were selected with probability proportional to PSU size and with independent selection in each sampling stratum within the sample allocation. A household listing obtained from each selected PSU before the main survey served as the sampling frame for the selection of sample households in the second stage. Thirty households were selected from each cluster, for a total sample size of 14,280 households (7,440 from urban and 6,840 from rural areas). Of the total households selected, interviews were successfully accomplished in 13786 households. A total of 15238 women from 14243 households were eligible, of whom 14845 women were successfully interviewed. Since cervical cancer screening and breast cancer screening are recommended after the age of 30, we only used data of 7,130 women who were aged 30 years and above for analysis.

### Data collection

Data collection for NDHS 2022 took place between January 5 and June 22, 2022.

### Dependent variables

**BCS uptake.** The woman was considered to uptake BCS if she reported that a doctor or other health care worker had ever tested her for breast cancer.

**CCS uptake.** The woman was considered to uptake CCS if she reported that a doctor or other healthcare worker had ever tested her for cervical cancer.

### Independent variables

The independent variables were selected based on the literature review and their availability in the NDHS 2022 datasets. The independent variables assessed in this study included ecological belt (mountains, hills, and Terai), place of residence (urban, rural), province (Koshi, Madhesh, Bagmati, Gandaki, Lumbini, Karnali, Sudurpaschim), age (in years), ethnicity (Brahmin or Chhetri, Dalit, Janajati, Madhesi, other), religion (Hindu, non-Hindu), marital status (unmarried, married or living together, divorced or non-living together), wealth quintile (poorest, poorer, middle, richer, richest), education (no education, basic, secondary, higher), occupation (not working, agriculture, professional or technical or manager or clerical, sales and service, skilled or unskilled labor, others), health insurance (covered, not covered), media exposure (present, not present), heard about cervical and breast cancer (yes, no), and heard about cervical and breast cancer screening (yes, no), parity (nullipara, primipara, multipara) and distance to health facility (big problem, not a big problem). The definition of independent variables is present in *S1 File*.

### Statistical analysis

We used R version 4.2.0 [24] and RStudio [25] for data cleaning and statistical analysis. We performed a weighted analysis using the "survey" package [26] to account for the complex survey design of NDHS 2022. Summary statistics for categorical variables were presented using frequency, percent, and 95% CI. Mean and 95% CI were presented to summarize numerical variables. We used univariable and multivariable logistic regression to determine the association of BCS and CCS with independent variables and reported crude odds ratio (COR) and adjusted odds ratio (AOR), and their 95% CI. A p-value of <0.05 was considered statistically significant.

### Ethical approval

We received approval from the DHS program to use the NDHS 2022 datasets. NHDS 2022 obtained ethical approval from the institutional review board of ICF International, United States of America (Reference number: 180657.0.001.NP.DHS.01, Date: 28th April 2022) and the ethical review board of Nepal Health Research Council (Reference number: 678, Date: 30th September 2021). Our analysis is based on a publicly available dataset of NDHS 2022. In the survey, written informed consent was obtained from all adult participants.

## Results

Table 1 illustrates the key characteristics of the women who participated in the NDHS 2022 survey. The age of the women ranged from 30 to 49 years with the mean age of 38.4 (95%CI: 38.2, 38.5) years. Of total 7,130 women, 41.5% aged 30–40 years and 58.5% aged 40–49 years. Majority of the women resided in urban areas (68.5%). Majority of the women were from the Terai region (53.7%) followed by the hilly regions (40.8%). In terms of the province, most women belonged to Bagmati (22.1%), followed by Madhesh (18.7%), Lumbini (18.2%), and Koshi (16.9%) provinces. In terms of ethnicity, the largest group was Janajati (38.1%), followed

**Table 1. Socio-demographic characteristics of Nepalese women aged 30–49 years (n = 7130).**

| Characteristic | All women aged 30–49 years, n = 7,130 | Women aged | |
| --- | --- | --- | --- |
| | | 30–40 years, n = 4,169 | 40–49 years, n = 2,961 |
| **Ethnicity** | | | |
| Brahmin/Chhetri | 2,112 (29.6) | 1,207 (29.0) | 904 (30.5) |
| Dalit | 978 (13.7) | 596 (14.3) | 382 (12.9) |
| Janajati | 2,719 (38.1) | 1,586 (38.0) | 1,133 (38.3) |
| Madheshi | 1,032 (14.5) | 600 (14.4) | 431 (14.6) |
| Others | 289 (4.1) | 179 (4.3) | 110 (3.7) |
| **Religion** | | | |
| Hindu | 5,961 (83.6) | 3,492 (83.8) | 2,468 (83.4) |
| Non-Hindu | 1,169 (16.4) | 677 (16.2) | 492 (16.6) |
| **Parity** | | | |
| Nullipara | 306 (4.3) | 230 (5.5) | 76 (2.6) |
| Primipara | 910 (12.8) | 685 (16.4) | 225 (7.6) |
| Multipara | 5,914 (82.9) | 3,254 (78.0) | 2,660 (89.9) |
| **Highest educational level** | | | |
| No education | 3,076 (43.1) | 1,364 (32.7) | 1,712 (57.8) |
| Basic | 2,143 (30.1) | 1,417 (34.0) | 726 (24.5) |
| Secondary | 1,561 (21.9) | 1,102 (26.4) | 459 (15.5) |
| Higher | 350 (4.9) | 286 (6.9) | 64 (2.1) |
| **Marital status** | | | |
| Unmarried | 109 (1.5) | 78 (1.9) | 30 (1.0) |
| Married or living together | 6,636 (93.1) | 3,933 (94.3) | 2,703 (91.3) |
| Divorced or not living together | 385 (5.4) | 158 (3.8) | 227 (7.7) |
| **Occupation** | | | |
| Not working | 1,194 (16.7) | 729 (17.5) | 464 (15.7) |
| Agriculture | 4,023 (56.4) | 2,197 (52.7) | 1,826 (61.7) |
| Professional/technical/managerial or clerical | 536 (7.5) | 372 (8.9) | 165 (5.6) |
| Sales and service | 723 (10.1) | 450 (10.8) | 273 (9.2) |
| Skilled or unskilled labor | 642 (9.0) | 418 (10.0) | 225 (7.6) |
| Other | 12 (0.2) | 4 (0.1) | 8 (0.3) |
| **Wealth index** | | | |
| Poorest | 1,189 (16.7) | 698 (16.7) | 492 (16.6) |
| Poorer | 1,381 (19.4) | 769 (18.4) | 612 (20.7) |
| Middle | 1,439 (20.2) | 828 (19.9) | 611 (20.6) |
| Richer | 1,459 (20.5) | 903 (21.7) | 556 (18.8) |
| Richest | 1,662 (23.3) | 971 (23.3) | 690 (23.3) |
| **Place of residence** | | | |
| Urban | 4,887 (68.5) | 2,891 (69.3) | 1,996 (67.4) |
| Rural | 2,243 (31.5) | 1,278 (30.7) | 965 (32.6) |
| **Ecological region** | | | |
| Mountain | 399 (5.6) | 224 (5.4) | 174 (5.9) |
| Hill | 2,906 (40.8) | 1,691 (40.6) | 1,215 (41.0) |
| Terai | 3,826 (53.7) | 2,254 (54.1) | 1,572 (53.1) |
| **Province** | | | |
| Koshi | 1,207 (16.9) | 689 (16.5) | 518 (17.5) |
| Madhesh | 1,336 (18.7) | 775 (18.6) | 561 (19.0) |

*(Continued)*

**Table 1.** (Continued)

| Characteristic | All women aged 30–49 years, | Women aged | |
|---|---|---|---|
| | n = 7,130 | 30–40 years, | 40–49 years, |
| | | n = 4,169 | n = 2,961 |
| Bagmati | 1,575 (22.1) | 922 (22.1) | 653 (22.0) |
| Gandaki | 728 (10.2) | 435 (10.4) | 294 (9.9) |
| Lumbini | 1,300 (18.2) | 785 (18.8) | 515 (17.4) |
| Karnali | 398 (5.6) | 234 (5.6) | 164 (5.6) |
| Sudurpaschim | 585 (8.2) | 329 (7.9) | 256 (8.6) |
| **Covered by health insurance** | 948 (13.3) | 544 (13.0) | 404 (13.6) |
| **Mass media exposure** | 3,597 (50.4) | 2,167 (52.0) | 1,429 (48.3) |
| **Distance to Health facility** | | | |
| Big problem | 2,884 (40.5) | 1,567 (37.6) | 1,318 (44.5) |
| Not a big problem | 4,246 (59.5) | 2,603 (62.4) | 1,643 (55.5) |

n: frequency; %: percent; CI: confidence interval

by Brahmin/Chhetri (29.6%), Madhesi (14.5%), and Dalit (13.7%). Approximately 93.1% of the women were married, while only 4.9% had attained higher education. Roughly half of the women were engaged in agriculture (56.4%), and around one-fifth were unemployed (16.7%). The women were distributed equally across all wealth quintiles. About half of the women reported exposure to any form of media on a weekly basis, 13.3% had health insurance coverage, and 40.5% perceived distance to health facility as a big problem.

Table 2 presents the awareness of and uptake of breast and cervical cancer screening among Nepalese women stratified by age group. Around three-fourths of the women had heard about breast cancer (77.6%) and cervical cancer (77.0%). Around half of the women had heard about breast cancer screening (52.8%) whereas around one-third (35.3) had heard about the cervical cancer screening. Among women aged 30–40 years and 40–49 years, the uptake of breast cancer screening in their lifetime was 6.0 and 7.1% whereas the uptake of cervical cancer screening was. 9.8% and 13.6% respectively.

Among women aged 30 to 49 years and who have heard about breast cancer, 68.1% (95% CI: 66.0, 70.1) had heard about BCS, and out of those who have heard about BCS, 11.0% (95% CI: 9.8, 12.5) had BCS at least once in a lifetime. Moreover, among women aged 30 years and above who have heard about cervical cancer, 45.8% (95%CI: 43.7, 48.0) had heard about CCS, and out of those who have heard about CCS, 26.8% (95%CI: 24.1, 29.7) had CCS at least once in a lifetime.

**Table 2. Heard of breast and cervical cancer and its screening uptake (n = 7,130).**

| Characteristics | 30–49 years women | | 30–40 years women | | 40–49 years women | |
|---|---|---|---|---|---|---|
| | N | % (95%CI) | n | % (95% CI) | n | % (95% CI) |
| Heard of Breast cancer | 5,535 | 77.6 (76.0, 79.2) | 3,256 | 78.1 (76.1, 79.9) | 2,279 | 77.0 (75.0, 78.8) |
| Heard of BCS | 3,768 | 52.8 (50.7, 55.0) | 2,247 | 53.9 (51.4, 56.4) | 1,521 | 51.4 (48.7, 54.0) |
| BCS uptake | 460 | 6.5 (5.72, 7.29) | 249 | 6.0 (5.12, 6.98) | 211 | 7.1 (5.99, 8.45) |
| Heard of Cervical Cancer | 5,491 | 77.0 (75.1, 78.8) | 3,241 | 77.7 (75.6, 79.8) | 2,251 | 76 (73.7, 78.2) |
| Heard of CCS | 2,517 | 35.3 (33.4, 37.3) | 1,465 | 35.1 (32.9, 37.4) | 1052 | 35.5 (33.0, 38.1) |
| CCS uptake | 811 | 11.4 (10.1, 12.8) | 408 | 9.8 (8.46, 11.3) | 403 | 13.6 (11.8, 15.6) |

n: frequency; %: percent; CI: confidence interval; BCS: Breast Cancer Screening; CCS: Cervical Cancer Screening

Among women aged 30–40 years and who have heard about breast cancer, 69.0% (95%CI: 66.6, 71.3) had heard about BCS, and out of those who had heard about BCS, 9.9% (95%CI: 8.6, 11.5) had BCS uptake at least once in their lifetime. Similarly, among women aged 30–40 years and hearing about cervical cancer, 45.2% (95%CI: 42.7, 47.7) had heard about CCS, and out of those who had heard about CCS, 23.0% (95%CI: 20.2, 26.1) had CCS uptake at least once in their lifetime.

Among women aged 40 to 49 years and hearing about breast cancer, 66.7% (95%CI: 64.0, 64.9) had heard about BCS, and out of those who had heard about BCS, 12.8% (95%CI: 10.7, 15.2) had BCS uptake at least once in their lifetime. Similarly, among women aged 40 to 49 years and hearing about cervical cancer, 46.7% (95%CI: 43.8, 49.7) had heard about CCS, and out of those who had heard about CCS, 32.1% (95%CI: 27.9, 36.5) had CCS uptake at least once in their lifetime.

Table 3 presents the association between breast and cervical cancer screening uptake among women aged 30 years and above. In a multivariable logistic regression to determine the association of BCS uptake with socio-demographic variables among women aged 30 years and above, it was observed that women from Terai had 0.54 times less likely to have BCS uptake compared to their counterparts from the mountainous region. The odds of BCS uptake were 2.08 times higher among Dalit women compared to Brahmin/Chhetri women. The odds of BCS uptake were 4% higher for each one-year increase in the age of a woman. The odds of BCS uptake were 1.49, 1.96, and 2.80 times higher in women with basic, secondary, and higher-level education, respectively, compared to women without formal education. Women who had heard about BCS had 6 times higher odds of BCS uptake. The odds of uptake of BCS among women engaged in agriculture was 31% lower compared to unemployed women.

In the multivariable analysis, the odds of CCS uptake were 24% lower in women from rural areas compared to urban areas. The odds of CCS uptake were 2.16 times higher among women from the Bagmati province and 2.09 times higher in Gandaki province, compared to women from the Koshi province. The odds of CCS uptake were 6% higher for each one-year increase in age of a woman. The odds of CCS uptake were 1.47 times higher among women with secondary-level education compared to women without formal education. The uptake was 8.61 times higher among women who had heard about CCS and 1.41 times higher among women with health insurance coverage compared to those who had not heard about CCS and did not have health insurance coverage, respectively.

## Discussion

This study assessed the association of socio-demographic factors with the uptake of BCS and CCS among women aged 30 years and above using data from NDHS 2022. Around three-fourths of the women had heard about breast cancer (78%) and cervical cancer (77%). Around half of the total women had heard about breast cancer screening whereas around one-fourth had heard about cervical cancer screening. Provinces, ethnicity, age, education, wealth, marital status, employment, media exposure, and health insurance coverage were identified as key factors associated with the uptake of BCS and CCS.

For CCS uptake, women from urban areas had higher uptake rates compared to rural areas. Women from certain provinces, such as Madhesh, Bagmati, Gandaki, and Karnali, had higher odds of CCS uptake than women from Koshi province. CCS uptake increased with age, education level, awareness of CCS, and health insurance coverage. Regarding BCS uptake, women from the mountain region had higher uptake rates than women from the Terai region. Dalit women showed higher odds of BCS uptake compared to Brahmin/Chhetri women. BCS uptake increased with age, education level, awareness of BCS, and marital status (married and

**Table 3. Factors associated with BCS and CCS uptake among Nepalese women aged 30 to 49 years.**

| Characteristics | BCS uptake | | | CCS uptake | | |
|---|---|---|---|---|---|---|
| | % (95%CI) | COR (95%CI) | AOR (95%CI) | % (95%CI) | COR (95%CI) | AOR (95%CI) |
| *Place of residence* | | | | | | |
| Urban | 7.4 (6.4, 8.6) | Ref | Ref | 13.8 (12.0, 15.7) | Ref | Ref |
| Rural | 4.4 (3.7, 5.2) | 0.57 (0.45, 0.73)** | 1.04 (0.79, 1.37) | 6.2 (5.4, 7.1) | 0.41 (0.33, 0.52)** | 0.76 (0.61, 0.96) * |
| *Ecological region* | | | | | | |
| Mountain | 6.3 (3.46, 11.2) | Ref | Ref | 9.8 (7.0, 13.5) | Ref | Ref |
| Hill | 8.6 (7.22, 10.2) | 1.4 (0.72, 2.72) | 0.75 (0.50, 1.15) | 14.6 (12.6, 17.0) | 1.58 (1.05, 2.38)* | 1.06 (0.65, 1.73) |
| Terai | 4.8 (4.03, 5.81) | 0.76 (0.39, 1.46) | 0.54 (0.31, 0.93)* | 9.1 (7.3, 11.2) | 0.92 (0.60, 1.42) | 0.99 (0.57, 1.73) |
| *Province* | | | | | | |
| Koshi | 5.1 (3.8, 6.9) | Ref | Ref | 7.3 (5.9, 9.0) | Ref | Ref |
| Madhesh | 3.8 (2.7, 5.40) | 0.73 (0.45, 1.19) | 1.31 (0.73, 2.35) | 6.8 (4.5, 10.1) | 0.93 (0.56, 1.52) | 1.93 (1.08, 3.43) |
| Bagmati | 10.8 (8.5, 13.6) | 2.25 (1.48, 3.40)** | 1.3 (0.78, 2.17) | 20.5 (16.6, 25.1) | 3.29 (2.32, 4.67)** | 2.16 (1.44, 3.23)** |
| Gandaki | 9.4 (7.1, 12.4) | 1.92 (1.23, 3.00)* | 1.24 (0.73, 2.09) | 15.9 (12.9, 19.5) | 2.42 (1.73, 3.38)** | 2.09 (1.40, 3.14)** |
| Lumbini | 5.4 (4.0, 7.2) | 1.05 (0.68, 1.64) | 1.07 (0.69, 1.66) | 8.7 (6.3, 11.8) | 1.21 (0.80, 1.83) | 1.19 (0.81, 1.74) |
| Karnali | 4.8 (3.6, 6.5) | 0.94 (0.60, 1.47) | 1.26 (0.75, 2.10) | 8.2 (5.9, 11.3) | 1.14 (0.75, 1.74) | 1.54 (0.96, 2.50) |
| Sudurpashchim | 3.3 (2.2, 5.0) | 0.64 (0.37, 1.08) | 0.98 (0.58, 1.67) | 8.2 (6.2, 10.8) | 1.14 (0.78, 1.67) | 1.19 (0.78, 1.82) |
| *Ethnicity* | | | | | | |
| Brahmin or Chhetri | 8.3 (7.0, 9.9) | Ref | Ref | 17.4 (14.9, 20.2) | Ref | Ref |
| Dalit | 6.1 (4.6, 8.1) | 0.72 (0.51, 1.01) | 2.08 (1.37, 3.16)** | 6.9 (5.0, 9.4) | 0.35 (0.25, 0.51)** | 0.79 (0.52, 1.19) |
| Janajati | 6.2 (5.0, 7.7) | 0.72 (0.53, 0.99)* | 1.21 (0.86, 1.70) | 10.8 (9.4, 12.5) | 0.58 (0.47, 0.71)** | 0.78 (0.59, 1.03) |
| Madheshi | 4.5 (3.1, 6.5) | 0.52 (0.33, 0.80)* | 1.46 (0.84, 2.53) | 7 (5.2, 9.5) | 0.36 (0.25, 0.52)** | 0.67 (0.42, 1.07) |
| Others | 3.6 (1.8, 6.8) | 0.41 (0.20, 0.83)* | 2.54 (1.07, 6.00) | 3.5 (1.5, 7.7) | 0.17 (0.07, 0.41)** | 0.4 (0.15, 1.08) |
| *Religion* | | | | | | |
| Hindu | 6.7 (5.9, 7.6) | Ref | Ref | 11.7 (10.3, 13.2) | Ref | Ref |
| Non-Hindu | 5.3 (4.0, 7.1) | 0.79 (0.57, 1.09) | 0.82 (0.56, 1.19) | 9.8 (7.8, 12.3) | 0.82 (0.63, 1.07) | 1.14 (0.79, 1.63) |
| *Age (in years)* | 38.0 (33.2, 44.0) [&] | 1.02 (1.00, 1.04)* | 1.04 (1.02, 1.07)** | 39.0 (35.0, 44.0) [&] | 1.04 (1.03, 1.06)** | 1.06 (1.04, 1.08)** |
| *Wealth index* | | | | | | |
| Poorest | 2.7 (2.0, 3.5) | Ref | Ref | 5.2 (4.1, 6.5) | Ref | Ref |
| Poorer | 4.0 (3.0, 5.4) | 1.54 (1.02, 2.33)* | 1.36 (0.88, 2.09) | 6.0 (4.7, 7.6) | 1.16 (0.81, 1.68) | 0.96 (0.65, 1.41) |
| Middle | 4.7 (3.58, 6.1) | 1.79 (1.20, 2.67)* | 1.46 (0.94, 2.29) | 7.3 (5.9, 8.9) | 1.44 (1.04, 1.99)* | 1.01 (0.68, 1.49) |
| Richer | 5.8 (4.6, 7.2) | 2.25 (1.56, 3.24)** | 1.32 (0.80, 2.15) | 10 (8.5, 11.8) | 2.05 (1.52, 2.76)** | 1.08 (0.72, 1.63) |
| Richest | 13.3 (11.1, 15.9) | 5.64 (3.98, 7.99)** | 1.69 (0.99, 2.89) | 25.1 (21.4, 29.3) | 6.16 (4.48, 8.47)** | 1.39 (0.87, 2.21) |
| *Education* | | | | | | |
| No education | 3 (2.4, 3.7) | Ref | Ref | 6 (5.1, 7.1) | Ref | Ref |
| Basic | 6.1 (5.0, 7.5) | 2.11 (1.51, 2.94)** | 1.49 (1.04, 2.13)* | 10.2 (8.6, 12.0) | 1.76 (1.37, 2.26)** | 1.24 (0.92, 1.66) |
| Secondary | 11.2 (9.3, 13.5) | 4.11 (3.05, 5.53)** | 1.96 (1.33, 2.88)** | 19.8 (16.5, 23.6) | 3.84 (2.90, 5.08)** | 1.47 (1.06, 2.04)* |
| Higher | 17.7 (12.8, 23.9) | 6.96 (4.46, 10.86)** | 2.80 (1.51, 5.19)** | 28.1 (21.7, 35.6) | 6.08 (4.17, 8.88)** | 1.4 (0.77, 2.55) |
| *Marital status* | | | | | | |
| Unmarried | 7.9 (3.49, 16.7) | Ref | Ref | 2.4 (0.4, 14.6) | Ref | Ref |
| Married or living together | 6.5 (5.7, 7.3) | 0.81 (0.34, 1.92) | 0.61 (0.16, 2.38) | 11.9 (10.5, 13.3) | 5.51 (0.79, 38.63) | 8.24 (1.03, 66.21)* |
| Divorced or not living together | 5.9 (3.8, 9.2) | 0.74 (0.30, 1.86) | 0.68 (0.16, 2.86) | 5.8 (3.8, 8.9) | 2.53 (0.34, 19.01) | 3.87 (0.44, 34.18) |
| *Occupation* | | | | | | |
| Not working | 10.5 (8.3, 13.1) | Ref | Ref | 16.7 (13.3, 20.6) | Ref | Ref |
| Agriculture | 3.8 (3.2, 4.6) | 0.34 (0.25, 0.47)** | 0.59 (0.42, 0.82)* | 7.1 (6.1, 8.1) | 0.38 (0.28, 0.52)** | 0.79 (0.56, 1.10) |
| Professional, manager, clerical | 13.3 (10.4, 16.8) | 1.31 (0.92, 1.85) | 0.75 (0.52, 1.08) | 25.7 (20.9, 31.2) | 1.73 (1.27, 2.35)** | 1.05 (0.73, 1.51) |
| Sales and service | 9.9 (7.4, 13.1) | 0.94 (0.64, 1.39) | 0.85 (0.58, 1.24) | 18 (14.4, 22.3) | 1.1 (0.79, 1.53) | 1.05 (0.75, 1.46) |
| Skilled/unskilled labor | 5.7 (3.7, 8.7) | 0.52 (0.30, 0.88) | 0.63 (0.37, 1.07) | 8.7 (6.4, 11.8) | 0.47 (0.31, 0.72)** | 0.7 (0.43, 1.11) |

*(Continued)*

**Table 3.** (Continued)

| Characteristics | BCS uptake | | | CCS uptake | | |
|---|---|---|---|---|---|---|
| | % (95%CI) | COR (95%CI) | AOR (95%CI) | % (95%CI) | COR (95%CI) | AOR (95%CI) |
| Other | 8.8 (1.5, 37.5) | 0.83 (0.13, 5.19) | 0.62 (0.11, 3.57) | 31.5 (4.6, 81.5) | 2.3 (0.25, 20.91) | 1.25 (0.22, 7.13) |
| *Covered by health insurance* | | | | | | |
| No | 5.7 (4.9, 6.5) | Ref | Ref | 9.9 (8.7, 11.3) | Ref | Ref |
| Yes | 11.7 (9.3, 14.7) | 2.21 (1.63, 2.99)** | 1.32 (0.97, 1.78) | 20.8 (17.4, 24.6) | 2.37 (1.85, 3.04)** | 1.41 (1.08, 1.83)* |
| *Mass media exposure* | | | | | | |
| No | 4.1 (3.4, 5.0) | Ref | Ref | 7.8 (6.6, 9.2) | Ref | Ref |
| Yes | 8.7 (7.6, 10.1) | 2.23 (1.73, 2.86)** | 1.42 (1.08, 1.86)* | 14.9 (13.0, 17.0) | 2.06 (1.67, 2.55)** | 1.2 (0.95, 1.53) |
| *Heard about BCS or CCS* | | | | | | |
| No | 1.3 (0.9, 1.8) | Ref | Ref | 3.0 (2.5, 3.6) | Ref | Ref |
| Yes | 11.0 (9.8, 12.5) | 9.33 (6.47, 13.45)** | 6.28 (4.42, 8.92)** | 26.8 (24.1, 29.7) | 11.96 (9.56, 14.96)** | 8.61 (6.84, 10.85)** |
| *Distance to HF* | | | | | | |
| Big problem | 4.8 (4.1, 5.8) | Ref | Ref | 7.3 (6.23, 8.6) | Ref | Ref |
| Not a big problem | 7.6 (6.5, 8.8) | 1.6 (1.26, 2.04)** | 0.82 (0.62, 1.08) | 14.1 (12.3, 16.2) | 2.08 (1.67, 2.60)** | 0.99 (0.77, 1.28) |
| *Parity* | | | | | | |
| Nullipara | 5.2 (2.9, 8.9) | Ref | Ref | 9.1 (5.8, 14.0) | Ref | Ref |
| Primipara | 9.9 (7.4, 13.2) | 2.02 (1.05, 3.92)* | 2.64 (1.06, 6.53)* | 15.5 (12.4, 19.1) | 1.83 (1.07, 3.13)* | 1.15 (0.59, 2.23) |
| Multipara | 6.0 (5.3, 6.8) | 1.17 (0.64, 2.13) | 2.45 (0.96, 6.25) | 10.9 (9.7, 12.2) | 1.22 (0.76, 1.97) | 1.27 (0.68, 2.37) |

* p-value<0.05

** p-value <0.001

& *Mean (95% confidence interval)*

%: percent; CI: confidence interval; COR: crude odds ratio; AOR: adjusted odds ratio; ref: reference group; BCS: Breast Cancer Screening; CCS: Cervical Cancer Screening

divorced women had higher odds compared to unmarried women). However, women engaged in agriculture had lower odds of BCS uptake compared to unemployed women.

Nepal consists of people from various ethnicities of which Brahmin/Chhetri are regarded as high caste or advantageous group, and Janajati and Dalit as lower or disadvantageous group. Previous studies suggest that women from the Dalit community compared to high caste groups are relatively easy to convince for health-seeking behavior despite their poor financial and social status [15, 27]. This might be the reason for the higher likelihood of BCS uptake among the Dalit group compared to Brahmin women. This could also be because health campaigns often tend to target disadvantaged communities over advantaged ones resulting in increased access and uptake of service among disadvantaged communities.

About 75% of women reported to have heard about breast cancer which falls within the range observed in various regions worldwide. Studies conducted in different countries have reported a wide range of breast cancer awareness among, typically varying from around 40% to 90% [28–34]. Awareness levels tend to be higher in more developed countries and regions with robust healthcare systems having access to greater information and better health services. The findings from a study conducted in Nepal demonstrated that around 54.3% women have heard about cervical cancer [35], while in Nigeria, Uganda, and India, several other studies reveal that the percentage of women who possess knowledge about cervical cancer ranged from 65% to 99% [36–38]. The relatively low awareness of breast cancer among women identified in our study underscores the urgent need for targeted awareness promotion, particularly among vulnerable and at-risk populations. Improving awareness of breast cancer is crucial in

fostering better screening habits and early detection, which are fundamental for enhancing the effectiveness of cancer treatment.

The findings of our study align with previous investigations conducted in Ethiopia, [39] Uganda, [40] and Lesotho [41] which have consistently shown that individuals who had access to media (television or radio) were more likely to be aware of BCS and CCS. This highlights the critical role of media information in improving awareness levels related to breast cancer. To maximize the reach and impact of health education efforts, it is crucial to utilize diverse and multiple channels to disseminate information about breast cancer. Both print and electronic media have proven to be effective tools in health education initiatives [41].

The present study's findings concerning BCS and CCS uptakes are consistent with prior research, corroborating the importance of knowledge acquisition in influencing screening utilization. The estimated uptake of CCS among women in Ethiopia, as reported by Kassie et al., was found to be 8% [42]. Moreover, their study highlights the significant association between knowledge of cervical cancer and the utilization of CCS [42]. A study examined the uptake of BCS in low-resource Asian countries and reported an overall utilization rate of 19% [43]. This study identified several factors associated with increased BCS uptake, including higher education levels, older age at first birth, female-headed households, access to media communication, and urban residency, in coherence with our study [43]. Similarly, Islam et al. conducted a systematic review, demonstrating that screening practices for both breast and cervical cancers were positively correlated with opportunities for knowledge acquisition [44]. Similar to our study, Islam, et al. also showed that women with higher education levels, urban residency, and employment outside the home exhibited increased screening behaviors [44]. These determinants reflect the complex interplay of sociocultural and economic factors that shape health-seeking behaviors, especially in resource-constrained regions. The findings highlight the importance of designing multifaceted interventions that address both knowledge gaps and socioeconomic barriers to improve breast cancer and cervical cancer screening rates.

### Strengths and limitations

This study has several strengths. First, this study is based on the nationally representative survey data so, the results of the study can be generalized for Nepal. Second, this study employed validated questionnaire and trained data enumerators to collect data. Third, we applied weighted analysis to account for non-response and complex survey design. Some limitations linked to this study included a) the directionality of the association of uptake of screening can't be established due to the cross-sectional nature of this study, b) uptake of BCS and CCS were self-reported which could be affected by recall-bias and social desirability.

### Implication to policymakers

In the context of Nepal, enhancing breast cancer screening awareness among women from different socio-demographic strata is pivotal to improving early detection and screening practices. Policymakers should prioritize health promotion interventions that target rural communities, women without formal education, and women from the poorest wealth quintile to raise the uptake of breast and cervical cancer screening. The findings from this study are valuable for clinicians and policymakers engaged in developing strategies for promoting breast and cervical cancer screening uptake.

### Conclusion

Though the significant number of women had heard of breast and cervical cancer and their screening, the uptake of BCS and CCS were relatively lower among Nepalese women. The

uptake of BCS was associated with ecological belt, occupation, ethnicity, education, and prior information on BCS whereas uptake of CCS was associated with place of residence, province, education, marital status, and enrollment in health insurance. The findings from this study emphasize the need for targeted awareness campaigns, continued efforts in health promotion and education, and comprehensive interventions to address these socio-demographic disparities and improve the awareness and uptake of breast and cervical cancer screenings. By tailoring strategies to address the identified factors, healthcare authorities can effectively increase awareness and uptake of screenings, leading to early detection and improved management of breast and cervical cancer cases.

## Supporting information

**S1 File. Operational definition of independent variables.**
(DOCX)

## Acknowledgments

We would like to acknowledge "The DHS Program" for providing us with the dataset for this study. In addition, we would like to acknowledge everybody who have directly or indirectly supported us throughout this study.

## Author Contributions

**Conceptualization:** Bipul Lamichhane, Bikram Adhikari, Lisasha Poudel, Achyut Raj Pandey.

**Data curation:** Bikram Adhikari, Bishnu Prasad Dulal.

**Formal analysis:** Bipul Lamichhane, Bikram Adhikari, Lisasha Poudel.

**Methodology:** Bipul Lamichhane, Santosh Giri.

**Supervision:** Bipul Lamichhane, Achyut Raj Pandey, Sampurna Kakchhapati, Bishnu Prasad Dulal, Ghanshyam Gautam, Sushil Chandra Baral.

**Validation:** Achyut Raj Pandey, Sampurna Kakchhapati, Santosh Giri, Bishnu Prasad Dulal.

**Writing – original draft:** Bipul Lamichhane, Lisasha Poudel, Saugat Pratap K. C.

**Writing – review & editing:** Bipul Lamichhane, Achyut Raj Pandey, Sampurna Kakchhapati, Saugat Pratap K. C., Santosh Giri, Bishnu Prasad Dulal, Deepak Joshi, Sushil Chandra Baral.

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
