## [Decision Letter · Decision Letter 0]

15 Nov 2023

PGPH-D-23-01693

Factors associated with uptake of breast and cervical cancer screening among Nepalese women: Evidence from Nepal demographic and health survey 2022

Dear Dr. Bipul Lamichhane

Thank you for submitting your manuscript to PLOS Global Public Health. After careful consideration, we feel that it has merit but does not fully meet PLOS Global Public Health’s publication criteria as it currently stands. Therefore, we invite you to submit a revised version of the manuscript that addresses the points raised during the review process.

We look forward to receiving your revised manuscript.

Kind regards,

Prabhdeep Kaur, DNB Medicine, MAE (Epidemiology)

Academic Editor

Journal Requirements:

Additional Editor Comments (if provided):

Reviewers' comments:

Reviewer's Responses to Questions

**Comments to the Author**

1. Does this manuscript meet PLOS Global Public Health’s publication criteria? Is the manuscript technically sound, and do the data support the conclusions? The manuscript must describe methodologically and ethically rigorous research with conclusions that are appropriately drawn based on the data presented.

Reviewer #1: Yes

Reviewer #2: Partly

2. Has the statistical analysis been performed appropriately and rigorously?

Reviewer #1: Yes

Reviewer #2: Yes

3. Have the authors made all data underlying the findings in their manuscript fully available (please refer to the Data Availability Statement at the start of the manuscript PDF file)?

Reviewer #1: Yes

Reviewer #2: Yes

4. Is the manuscript presented in an intelligible fashion and written in standard English?

Reviewer #1: Yes

Reviewer #2: Yes

5. Review Comments to the Author

Reviewer #1: The survey is an important exercise and a necessary initiative towards common cancer control in never screened population.The study article is comprehensive with a good representative sample size with analysis of relevant demographic variables. The manuscript is well scripted with robust data inputs.

The authors may explain the following:

1.Around one-third of women respondents being less than 20 years of age, has HPV vaccination been included in the questionnaire?

2.Has the authors explored the reason behind dalit women having better attitude towards screening uptake than the socially considered upper class of brahmin/chetri women?

Reviewer #2: Thank you for asking me to review your paper, which I enjoyed reading. My main concern is with the age group of women who are being asked about screening, given that cervical screening is not recommended to commence before age 30 in most settings and breast screening typically at 40-50 years. I think the analysis and interpretation would really benefit from considering this, particularly in relation to reported cancer screening uptake. Are you also able to provide more information about who provides screening services in Nepal and what their capacity is (i.e. if more women want to screen is it possible? Where are these services located? Is it affordable to most people? Is awareness and acceptance the major barrier to screening or is the availability of and access to services also a major issue?)

Intro

Para one

Line one, Fix wording – insert ‘a’ before major and replace ‘the’ with ‘a’ leading. ie Cancers are a major… and a leading cause of death. Last sentence – 90% of which new cases and deaths – from cervical cancer or something else? Specify.

Third paragraph reword ‘Breast and cervical cancers screening is a crucial strategy’ to ‘Breast and cervical cancer screening are crucial strategies….’

Line 6 – save lives not saves lives

Line6/7 ‘preserve quality of life’ not ‘ preserves the quality of life’

Para 4 last line

Reword ‘and can inform policymakers to assist in the development of policies and targeted interventions to improve screening uptake.’

Methods

Second para line 2, consisting of 53

Line 4 – ‘of whom’, not ‘of which’ (same word replacement needed, second last line of sampling section also)

Line 5 typo delete 1 from 148.9 % male

Dependent variables: unclear – do these mean if the woman ever reported that a doctor or HCW had tested her? This is what I assume from the limitations section saying it was self reported. Was there any explanation to participants about what being screened for either of these conditions would entail (eg breast examination or speculum use etc)? How do we know that the responses are valid/likely to indicate that screening occurred?

Results

What age groups are screen eligible in Nepal for each program? Noting that the sample comprises women aged 15-49 years and that the median age is 29, it is worth noting that the general start age for cervical screening per WHO guidelines is 30 and for breast screening in many countries it is 50. So you would not expect most of these women to be screening even if they were aware of the programs and the programs were available. I suggest that the analysis should be redone or at least stratified by those who are age eligible for the programs rather than including younger women? You have documented in your results a very strong association between screening and age as would be expected for programs where younger people should not be screening.

6. PLOS authors have the option to publish the peer review history of their article (what does this mean?). If published, this will include your full peer review and any attached files.

**Do you want your identity to be public for this peer review?** For information about this choice, including consent withdrawal, please see our Privacy Policy.

Reviewer #1: **Yes: **MALLIGA J SUBRAMANIAN

Reviewer #2: **Yes: **Julia Brotherton

---

## [Decision Letter · Decision Letter 1]

6 Feb 2024

Factors associated with uptake of breast and cervical cancer screening among Nepalese women: Evidence from Nepal Demographic and Health Survey 2022

PGPH-D-23-01693R1

Dear Bipul Lamichhane,

We are pleased to inform you that your manuscript 'Factors associated with uptake of breast and cervical cancer screening among Nepalese women: Evidence from Nepal Demographic and Health Survey 2022' has been provisionally accepted for publication in PLOS Global Public Health.

Best regards,

Prabhdeep Kaur, DNB Medicine, MAE (Epidemiology)

Academic Editor

Reviewer Comments (if any, and for reference):

Reviewer's Responses to Questions

**Comments to the Author**

1. If the authors have adequately addressed your comments raised in a previous round of review and you feel that this manuscript is now acceptable for publication, you may indicate that here to bypass the “Comments to the Author” section, enter your conflict of interest statement in the “Confidential to Editor” section, and submit your "Accept" recommendation.

Reviewer #1: All comments have been addressed

Reviewer #2: (No Response)

2. Does this manuscript meet PLOS Global Public Health’s publication criteria? Is the manuscript technically sound, and do the data support the conclusions? The manuscript must describe methodologically and ethically rigorous research with conclusions that are appropriately drawn based on the data presented.

Reviewer #1: Yes

Reviewer #2: Yes

3. Has the statistical analysis been performed appropriately and rigorously?

Reviewer #1: Yes

Reviewer #2: Yes

4. Have the authors made all data underlying the findings in their manuscript fully available (please refer to the Data Availability Statement at the start of the manuscript PDF file)?

Reviewer #1: Yes

Reviewer #2: Yes

5. Is the manuscript presented in an intelligible fashion and written in standard English?

Reviewer #1: Yes

Reviewer #2: Yes

6. Review Comments to the Author

Reviewer #1: Abstract: Page 5 Lines 25-27 Women from Mountain or Terrai-who have better uptake of screening? Please clarify. Conflicting report with the results: page 13 first paragraph first 3 lines

Page 6, line 83 recommendation to replace recommend Page 7 Line 89 –rephrase as access to screening , word repeated

Page 7, Line 125 – biomarker specialist? Please clarify

Page 12, Lines 197, 202 CSS Typo To replace CSS with BCS

Page 13, Table 3 Age in years – not shown –incomplete data

Page 15, Discussion: Lines 28-30 –Provide references for the two studies in Nepal that showed around 77% awareness level

Page 16 Policy implication: Lines 84,85 The authors can rephrase the sentence to target all strata of women, since the authors explain that the health campaigns have already increased uptake among disadvantaged population like Dalit women.

Reviewer #2: Thank you for addressing my previous comments and reanalysing for those aged 30 and over only. I note a couple of errors in the redrafting for correction below.

Delete this sentence line 237-8

A total of 4.4% and 6.4% women 238 had breast and cervical cancer screening in lifetime.

Correction line 260-1 (CCS not BCS before 26.8%)

Moreover, among women aged 30 years and above who have heard about cervical

260 cancer, 45.8% (95%CI: 43.7, 48.0) had heard about CCS, and out of those who have heard about BCS,

261 26.8% (95%CI: 24.1, 29.7) had CCS at least once in a lifetime.

Same error lines 263 and 268 where should be BCS not CCS

7. PLOS authors have the option to publish the peer review history of their article (what does this mean?). If published, this will include your full peer review and any attached files.

**Do you want your identity to be public for this peer review?** For information about this choice, including consent withdrawal, please see our Privacy Policy.

Reviewer #1: **Yes: **malliga j subramanian

Reviewer #2: **Yes: **Julia Brotherton
